# Mice lacking the mitochondrial exonuclease MGME1 develop inflammatory kidney disease with glomerular dysfunction

**Dusanka Milenkovic**[1], **Adrián Sanz-Moreno**[2], **Julia Calzada-Wack**[2], **Birgit Rathkolb**[2,3,4], **Oana Veronica Amarie**[2], **Raffaele Gerlini**[2,4], **Antonio Aguilar-Pimentel**[2], **Jelena Misic**[5], **Marie-Lune Simard**[1], **Eckhard Wolf**[3], **Helmut Fuchs**[2], **Valerie Gailus-Durner**[2], **Martin Hrabě de Angelis**[2,4,6]*, **Nils-Göran Larsson**[5]*

1 Max Planck Institute for Biology of Ageing, Cologne, Germany, 2 Institute of Experimental Genetics, German Mouse Clinic, Helmholtz Zentrum München, German Research Center for Environmental Health GmbH, Neuherberg, Germany, 3 Institute of Molecular Animal Breeding and Biotechnology, Gene Center, Ludwig-Maximilians-University Munich, Munich, Germany, 4 German Center for Diabetes Research (DZD), Neuherberg, Germany, 5 Department of Medical Biochemistry and Biophysics, Karolinska Institutet, Stockholm, Sweden, 6 Chair of Experimental Genetics, TUM School of Life Sciences, Technische Universität München, Freising, Germany

* nils-goran.larsson@ki.se (NGL); hrabe@helmholtz-muenchen.de (MH)

**Data Availability Statement:** All relevant data are within the manuscript and its Supporting Information files.

## Abstract

Mitochondrial DNA (mtDNA) maintenance disorders are caused by mutations in ubiquitously expressed nuclear genes and lead to syndromes with variable disease severity and tissue-specific phenotypes. Loss of function mutations in the gene encoding the mitochondrial genome and maintenance exonuclease 1 (MGME1) result in deletions and depletion of mtDNA leading to adult-onset multisystem mitochondrial disease in humans. To better understand the *in vivo* function of MGME1 and the associated disease pathophysiology, we characterized a *Mgme1* mouse knockout model by extensive phenotyping of ageing knockout animals. We show that loss of MGME1 leads to *de novo* formation of linear deleted mtDNA fragments that are constantly made and degraded. These findings contradict previous proposal that MGME1 is essential for degradation of linear mtDNA fragments and instead support a model where MGME1 has a critical role in completion of mtDNA replication. We report that *Mgme1* knockout mice develop a dramatic phenotype as they age and display progressive weight loss, cataract and retinopathy. Surprisingly, aged animals also develop kidney inflammation, glomerular changes and severe chronic progressive nephropathy, consistent with nephrotic syndrome. These findings link the faulty mtDNA synthesis to severe inflammatory disease and thus show that defective mtDNA replication can trigger an immune response that causes age-associated progressive pathology in the kidney.

## Author summary

We have addressed the controversy of the role of the mitochondrial genome and maintenance exonuclease 1 (MGME1) in mtDNA metabolism by characterization of knockout

**Funding:** This work was supported by grants to NGL from the Swedish Research Council (2015-00418), Swedish Cancer Foundation (2021.1409), the Knut and Alice Wallenberg foundation, European Research Council (ERC Advanced Grant 2016-741366), Novo Nordisk Foundation (NNF20OC0063616), Diabetesfonden (DIA2020-516) and grants from the Swedish state under the agreement between the Swedish government and the county councils (SLL2018.0471). GMC is supported by grants from the German Federal Ministry of Education and Research (Infrafrontier grant 01KX1012 to MHdA) and the German Center for Diabetes Research (DZD) (MHdA). The funders had no role in study design, data collection and analysis, decision to publish, or preparation of the manuscript.

**Competing interests:** I have read the journal's policy and the authors of this manuscript have the following competing interests. NGL is a scientific founder and holds stock in Pretzel Therapeutics, Inc. The other authors have no competing interests.

mice. Our findings show that loss of MGME1 leads to increased *de novo* formation of linear deleted mtDNA, thus contradicting previous report that MGME1 degrades long linear mtDNA molecules. In addition, we report that loss of MGME1 leads to age-associated pathology manifested as progressive weight loss, cataract and retinopathy. Aged knockout mice also develop kidney inflammation leading to glomerular changes, fibrosis and nephrotic syndrome. Defective mtDNA replication causing the formation of linear deleted mtDNA can thus trigger an immune response that leads to the development of progressive kidney disease in ageing animals.

## Introduction

Impaired replication or maintenance of mitochondrial DNA (mtDNA) lead to mitochondrial diseases, a clinically and genetically heterogeneous group of multisystemic disorders affecting various organs [1,2]. Defects in mtDNA can be either quantitative, causing mtDNA depletion, or qualitative, causing accumulation of deletions and/or point mutations of mtDNA. The expression of mtDNA is completely dependent on nuclear genes that encode proteins that are synthesized in the cytosol and imported into the mitochondrial matrix [3]. At least two hundred nucleus-encoded proteins are needed for maintenance, replication and transcription of mtDNA, as well as biogenesis of mitochondrial ribosomes [4]. The basic components of the mtDNA replication machinery are known and mutations in the catalytic and accessory subunits of mitochondrial DNA polymerase (*POLγA* and *POLγB*) [5, 6] the replicative DNA helicase (*TWNK*) [7], the mitochondrial single-stranded DNA binding protein (*SSBP1*) [8] and the mitochondrial genome and maintenance exonuclease 1 (*MGME1*) [9,10], cause mutations and/or depletion of mtDNA, which, in turn, impair mitochondrial function and cause mitochondrial disease syndromes. According to one model supported by biochemical data, MGME1 is a mitochondrial nuclease that processes newly replicated 5' DNA ends to facilitate ligation when mtDNA synthesis is completed [9,11–13]. Based on studies in cell lines, an additional function for MGME1 in degradation of long linear mtDNA fragments was proposed [14]. Both models predict that the absence of MGME1 will lead to the formation of long linear deleted mtDNA molecules, but due to different mechanisms, i.e. increased formation or decreased degradation, respectively.

Loss of function mutations in the *MGME1* gene cause human disease syndromes with mtDNA depletion and accumulation of mtDNA rearrangements (MIM#615084) [9,10]. Affected patients develop a range of symptoms in various organs, including brain, skeletal muscle, heart and gastrointestinal organs [9,10]. The majority of the patients develop adult onset disease, but onset in childhood has also been described [10]. We recently generated *Mgme1* knockout mice and found that they had prominent mtDNA replication aberrations manifested as replication stalling, mtDNA depletion, formation of long linear deleted molecules and an increase of short single stranded DNA products caused by prematurely aborted replication. In our initial characterization, we established that *Mgme1* knockout mice were born at Mendelian proportions and had no obvious changes in gross appearance up until one year of age [13]. Consistent with our results, the international mouse phenotyping consortium project (https://www.mousephenotype.org/data/genes/MGI:1921778#section-associations) revealed only minor changes in an independent *Mgme1* knockout mouse strain analyzed at the age of 10 weeks. Because humans with *MGME1* mutations typically develop late-onset-mitochondrial disease [9,12], we decided to characterize ageing cohorts of *Mgme1* knockout mice by an extensive phenotypic analysis, including clinical chemistry (haematology, metabolism and organ function), energy metabolism (indirect calorimetry, body composition),

evaluation of different organ systems, immune system characterization and pathological assessment of tissue changes. We report here that *Mgme1* knockout mice show reduced weight gain during aging and even weight loss later in life. In addition, the *Mgme1* knockout mice develop cataracts and retinopathy at 65–70 weeks of age. Strikingly, aged mice also develop kidney inflammation, glomerular changes and chronic progressive nephropathy with albuminuria, and die prematurely at ~70 weeks of age. Our findings provide a direct link between defective mtDNA replication and ageing-associated inflammation.

## Results

### Mgme1 knockout mice have a shorter life span

Cohorts of wild-type and *Mgme1*$^{-/-}$ animals of both sexes were aged, and their body weights were monitored every four weeks starting from eight weeks of age (Figs 1 and S1). We initially planned to perform the standard late-adult screening pipeline by phenotyping *Mgme1* knockout mice from the age of 71 weeks and onwards (www.mouseclinic.de) [15]. However, as the *Mgme1*$^{-/-}$ mice aged, we noticed that animals started to die prematurely from ~50 weeks of age (Figs 1A and S1A). The cause of death could not be determined at that time because the animals were found dead in their cages without previous obvious deterioration of their general condition. Because of the high attrition rate at the age of 68 weeks, we decided to start the examination of the ageing *Mgme1*$^{-/-}$ mice earlier than initially planned. Due to animal welfare reasons, we decided to perform a modified phenotyping protocol with combined neurological and morphological examinations, cardiovascular monitoring, body composition measurements and a final blood sampling followed by pathological analyses, as described in S1 Table.

### MGME1 deficiency results in body weight decline

Both female and male *Mgme1*$^{-/-}$ animals showed progressive weight loss (Fig 1B). We detected a significantly decreased body weight compared to controls in male *Mgme1*$^{-/-}$ animals from week 28, and in females *Mgme1*$^{-/-}$ from week 52 (Fig 1B). We performed non-invasive qNMR scans to determine body composition and noted significantly decreased fat content in percent of body weight and decreased adiposity index in male *Mgme1*$^{-/-}$ mice compared to wild-type controls (Figs 1C and S1B). Conversely, the lean mass proportion of male knockout animals was significantly increased (S1B Fig).

### MGME1 is not essential for degradation of linear mtDNA fragments

In addition to its established role in the processing of DNA flaps, MGME1 has been suggested to act together with POLγA and TWINKLE as a component of an enzymatic machinery that can degrade linear mtDNA fragments [14,16,17]. Different tissues of *Mgme1*$^{-/-}$ mice contain substantial amounts of linear subgenomic fragments [13]. If impaired degradation explains the presence of these deleted molecules, they would be predicted to accumulate over time. Using Southern blot analysis, we quantified the amount of linear deleted molecules relative to total mtDNA in young (~10 weeks) and old animals (~50 weeks), but found no accumulation of the linear deleted mtDNA with age in liver, kidney, heart, skeletal muscle and brain (Figs 2A and S2A–S2E) of *Mgme1*$^{-/-}$ mice. These findings argue that the persistence of linear deleted mtDNA is not due to deficient degradation, but is rather explained by a constant formation of deleted molecules due to defective mtDNA replication. To further investigate this issue, we performed *in organello* mtDNA replication experiments with pulse/chase labelling to visualize both formation and stability of linear deleted mtDNA in the absence of MGME1 (Fig 2B). With this experimental setup, we monitored *de novo* formation of linear deleted mtDNA

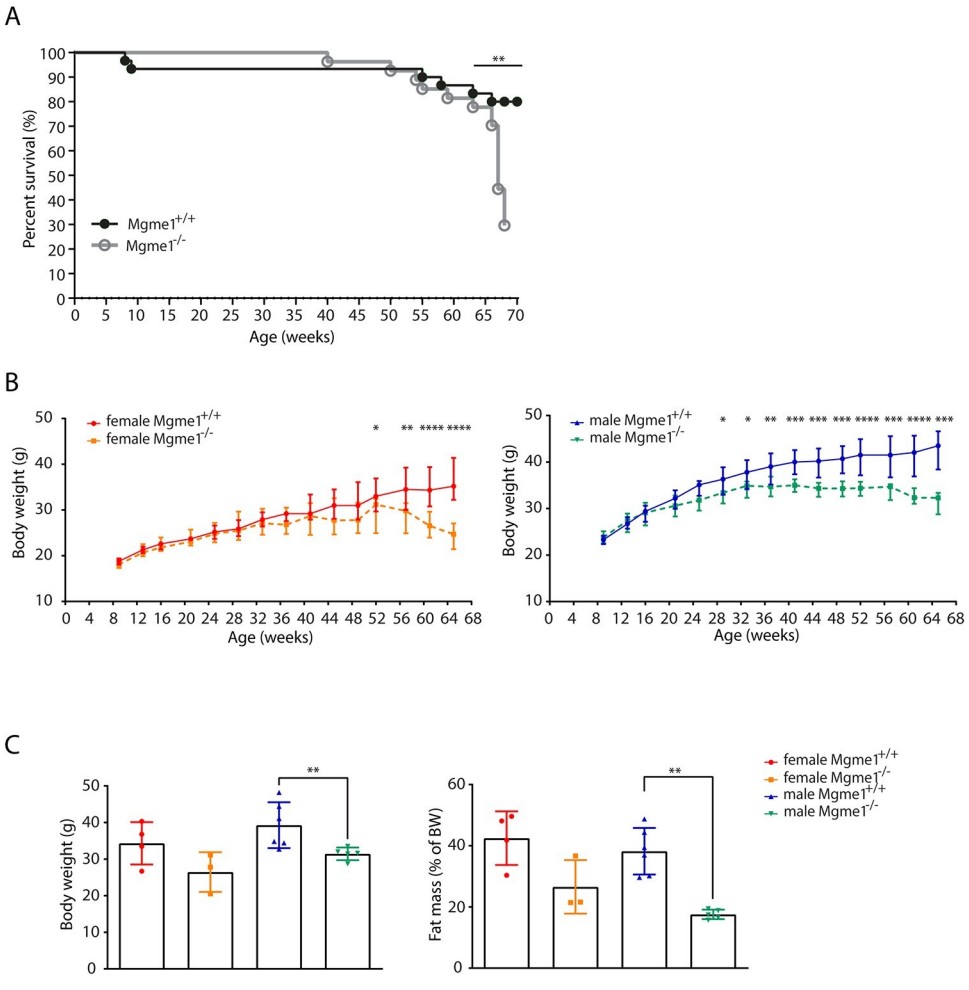

**Fig 1. MGME1 deficiency leads to shorter life span in mice.** (A) High attrition rate was observed in older *Mgme1*$^{-/-}$ animals. p = 0.0008 Log-rank (Mantel-Cox) test. 19 from 27 mutant and 6 from 30 control animals did not reach the end of phenotyping protocol. (B) *Mgme1* knockout mice do not gain weight as they age compared with controls and show progressive weight loss at the later time points, * P ≤ 0.05, ** P ≤ 0.01, *** P ≤ 0.001, **** P ≤ 0.0001 Unpaired Mann-Whitney Test. (C) Body weight measured at 68 weeks of age of n = 4 wild-type and 3 *Mgme1*$^{-/-}$ females and 6 wild-type and 5 *Mgme1*$^{-/-}$ males. Fat mass is indicated as percent of body weight. Values are given as mean ± SD, ** P ≤ 0.01. Unpaired Mann-Whitney Test.

fragments and found that they were degraded although MGME1 was absent. The findings in ageing *Mgme1*$^{-/-}$ mice (Fig 2A) and characterization of *in organello* mtDNA replication in mitochondria lacking MGME1 (Fig 2B) thus show that increased formation of linear deleted mtDNA occurs when MGME1 is absent. In addition, Southern blot analyses of full-length mtDNA and total mtDNA (full-length + linear deleted mtDNA) in heart and kidney of *Mgme1*$^{-/-}$ mice revealed clear mtDNA depletion accompanied by increased levels of 7S DNA (Figs 2C and 2D, S3A and S3B). However, also these phenotypes were not aggravated with age, consistent with mtDNA replication defect in the *Mgme1*$^{-/-}$ mice.

## Altered lens and retina morphology in eye tissue of Mgme1$^{-/-}$ mice

Histological analysis of eye tissue sections from *Mgme1*$^{-/-}$ mice showed alterations of the lens matrix structure (Fig 3A and 3B). In wild-type lenses, the epithelial cells are arranged in a

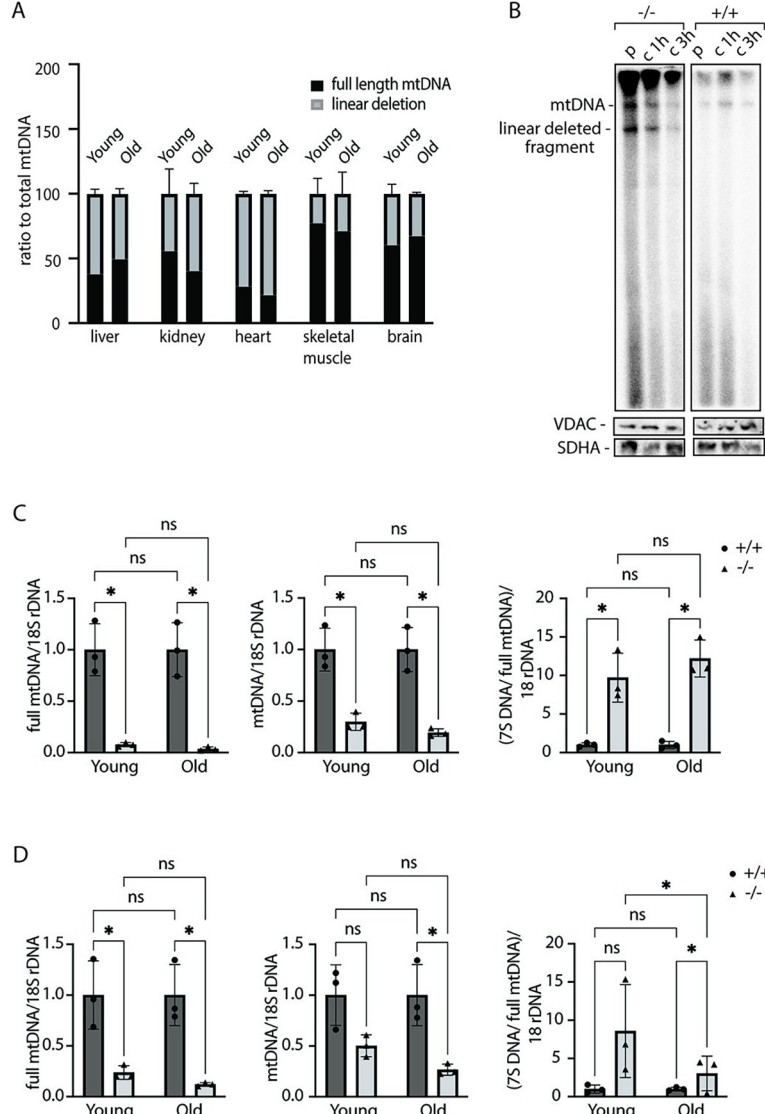

**Fig 2. Linear deleted subgenomic fragments do not accumulate with age in *Mgme1*⁻/⁻ mice.** (A) Southern blot quantification of linear deletion fragment signal in ratio to total mtDNA signal in liver, kidney,heart, skeletal muscle and brain of *Mgme1*⁻/⁻ mice at 10 and 50 weeks of age. Error bars represent SD. (B) *De novo* DNA synthesis in heart mitochondria isolated from control (+/+) and Mgme1 knockout (-/-) mice. Mitochondria were pulse labeled (p) for 2 h and the chase was performed for 1 and 3 h. SDHA and VDAC levels on western blot represent loading control for the input mitochondria. (C) Quantification of full length mtDNA (left panel), total mtDNA (middle panel) and 7S DNA (right panel) in wild-type and MGME1 knock-out heart tissue of young (10 weeks) and old (55 weeks) animals. Values are given as mean ± SD. * P ≤ 0.05, T-test. (D) As (C) but in kidney tissue.

monolayer under the lens capsule without gaps or derangement of adjacent fiber cells. The lens matrix of the *Mgme1*⁻/⁻ mice contained migrated epithelial cells and the normally densely packed and regularly arranged fiber cells in the lens matrix were swollen and abnormally arranged (Fig 3C). All retinal layers were present in *Mgme1* knockout mice, but the total retinal thickness as well as the thickness of each of the three retinal layers, i.e., outer plexiform layer (OPL), inner nuclear layer (INL) and outer nuclear layer (ONL) near the optic nerve was reduced in the *Mgme1*⁻/⁻ mice (Fig 3D and 3E).

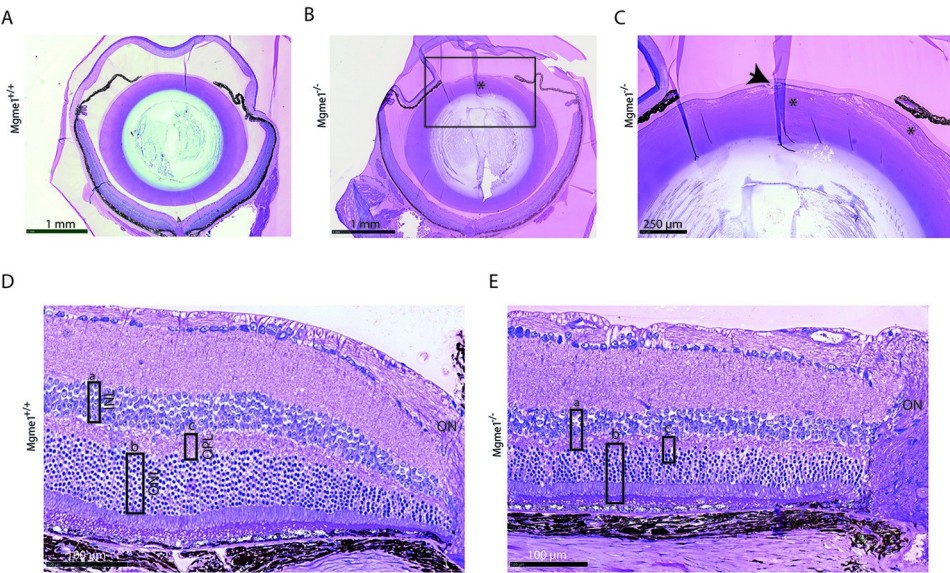

**Fig 3. *Mgme1*<sup>-/-</sup> eye tissue histology images show clear alterations of lens and retina morphology.** (A) Sections through the *Mgme1*<sup>+/+</sup> and (B) the *Mgme1*<sup>-/-</sup> eye, indicate at first glance alterations of the lens matrix structure (*); (C) enlarged *Mgme1*<sup>-/-</sup> anterior lens image reveals alterations of the lens fiber cells: they are swollen and disorganized (*), affecting the stability of the whole lens matrix. Additionally changes are present as to the epithelial cell monolayer, located underneath the lens capsule, shown here as an accumulation of epithelial cells that seem to make place for invaginations of the lens capsular material (arrow). (D) Retinal layers near ON images show an arbitrary thickness evaluation of some of the layers as rectangles (rectangle a—INL, rectangle b—ONL, rectangle c—OPL) in their 'control' size that in E. are placed over the *Mgme1*<sup>-/-</sup> retinal layers, to indicate the reduction in all three layers. Abbreviations: INL—inner nuclear layer, ONL—outer nuclear layer, OPL—outer plexiform layer, ON—optic nerve.

## MGME1 deficiency causes kidney inflammation and nephropathy

Measurement of cytokines in plasma revealed increased levels of IL-6, TNF-α, Kc/Gro, IL-2, IL-10 and IFN-γ in *Mgme1*<sup>-/-</sup> knockout animals (Fig 4), consistent with ongoing inflammation. The results of the clinical chemistry screening revealed marked increases of plasma creatinine and urea levels in ageing *Mgme1*<sup>-/-</sup> mice, indicative of a severely impaired renal function (Fig 5A). Urea and creatinine as well as cholesterol levels were significantly increased in male mutants (Wilcoxon Rank Sum Test p = 0.004), whereas female mutants showed a non-significant trend towards an increase. The plasma albumin concentration was lower in mutant mice of both sexes (Wilcoxon Rank Sum Test, over all genotype effect p = 0.036). The slightly decreased plasma albumin and clearly increased plasma cholesterol concentrations represented typical alterations that hint towards significant renal protein loss, as seen in nephrotic syndrome (Fig 5A, S2 Table). In addition, we found altered electrolyte levels in mutant animals with elevated sodium and potassium levels (WRST over all genotype effect p = 0.04 and p = 0.014, respectively) and decreased chloride concentration in males (WRST males p = 0.004). The calcium/phosphate (Ca, Pi) ratio was increased in mutant animals, due to increased calcium levels (WRST over all p = 0.003), and this alteration was associated with significantly increased alkaline phosphatase activity (WRST over all p = 0.001) consistent with significant effects on mineral balance with effects on bone metabolism in mutant animals (S2 Table). The nephrotic syndrome is typically caused by damage of the kidney glomeruli and we therefore performed a more detailed pathohistological analysis of the kidney. Lesions compatible with chronic progressive nephropathy, a common age-related disease in rodents [18], were obvious in all 68-week-old male (5 out of 5) and most female (2 out of 3) *Mgme1*<sup>-/-</sup> mice

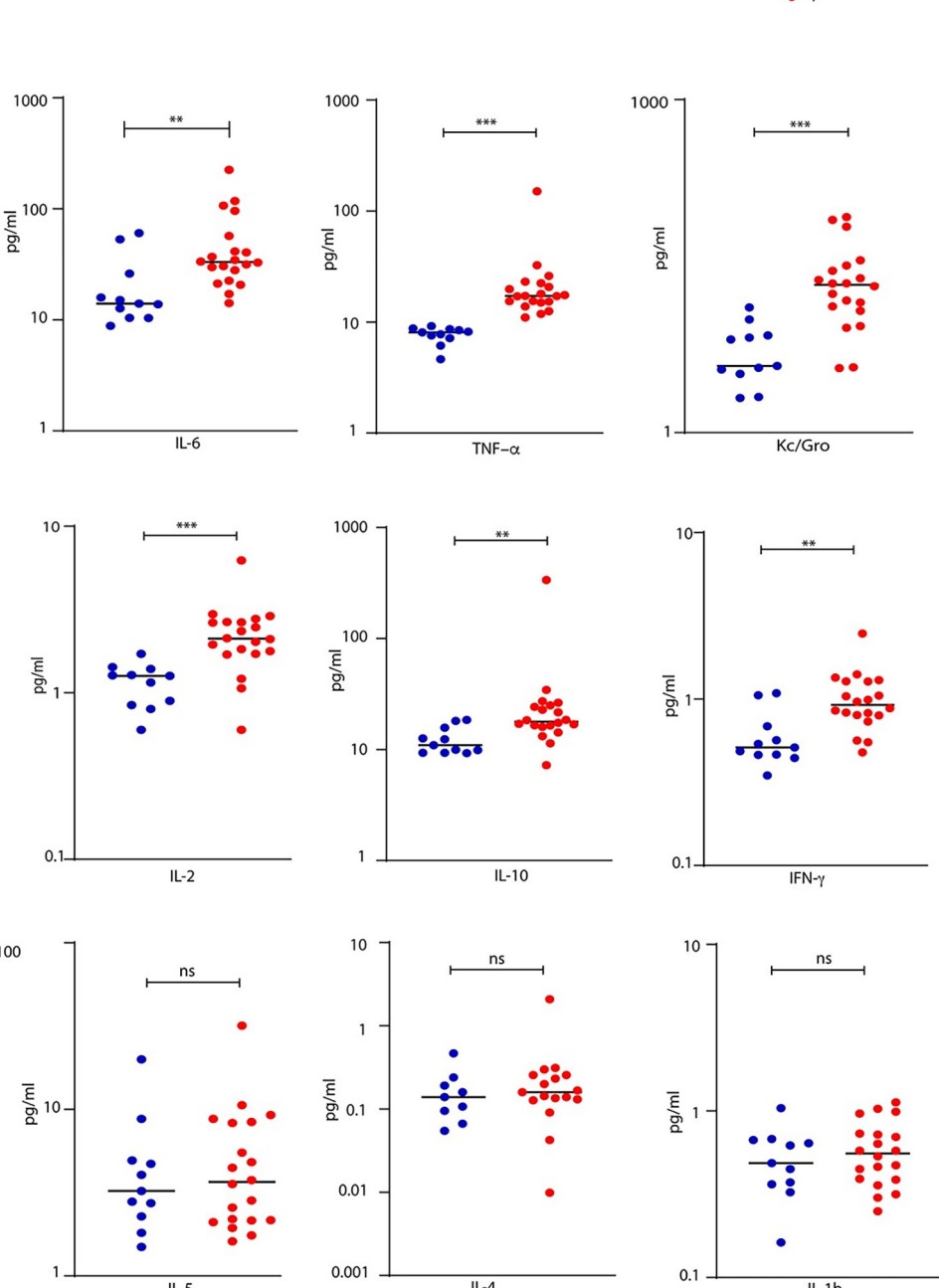

**Fig 4. Measurement of cytokines. Multiplex measurement of the proinflammatory cytokines IL-6, TNF-α, KC/ GRO, IL-2, IL-10, IFN-γ, IL-5, IL-4 and IL-1b was performed in plasma of control and *Mgme1*<sup>-/-</sup> mice.** Aged *Mgme1*<sup>-/-</sup> mice accumulate several proinflammatory cytokines in plasma.

analyzed at the end of the GMC pipeline (Fig 5B–5D). This pathology was also observed in hematoxylin and eosin (H&E) stainings of renal sections from all prematurely dead *Mgme1*<sup>-/-</sup> mice where histology was performed (3 females and 2 males). No signs of chronic progressive nephropathy were detected in any of the 10 control animals studied (4 females and 6 males). H&E and periodic acid-Schiff (PAS) stainings revealed glomerular changes with mesangial

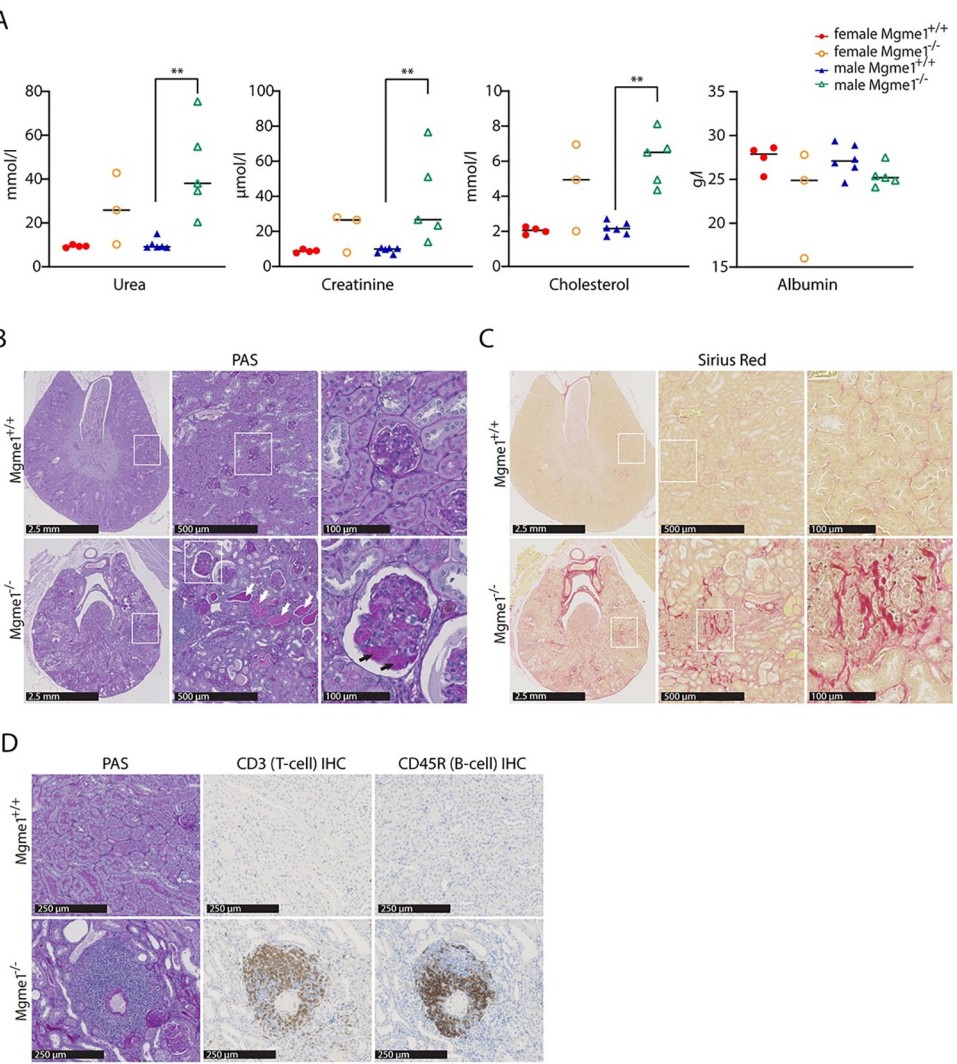

**Fig 5. *Mgme1*<sup>-/-</sup> mice display altered renal function and a severe nephropathy** (A) Plasma urea, creatinine, cholesterol and albumin levels of *ad libitum* fed mice at 68 weeks of age; n = 4 WT and 3 *Mgme1*<sup>-/-</sup> females and 6 WT and 5 *Mgme1*<sup>-/-</sup> males. The median is shown as a line. ** P ≤ 0.01 (B) Representative pictures of PAS (Periodic acid-Schiff) stainings of kidney sections from control and *Mgme1*<sup>-/-</sup> mice. Multifocal hyaline proteinaceous casts (indicated with white arrows) and lymphocytic immune infiltrates (indicated with an asterisk) were found in the kidneys of *Mgme1*<sup>-/-</sup> animals (lower central panel). PAS-positive mesangial matrix expansion (indicated with black arrows) was obvious in many glomeruli of knockout mice as shown at higher magnification (lower panel, right). (C) Representative pictures of Sirius Red stainings at different magnifications show pronounced tubulointerstitial fibrosis in the kidneys of knockout mice. (D) Immune infiltrates in the kidneys of *Mgme1*<sup>-/-</sup> animals were composed mainly by T and B cells, as demonstrated by CD3 and CD45R/B220 immunohistochemistry, respectively.

expansion, abundant hyaline proteinaceous casts and lymphocytic immune infiltrates in the kidneys of *Mgme1*<sup>-/-</sup> mice (Fig 5B). Generalized tubulointerstitial fibrosis was also evident in knockout mice after Sirius Red staining (Fig 5C). A portion of the glomeruli from knockout animals was also positive in the Sirius Red staining, pointing to a partial glomerulosclerosis. A more detailed characterization of the renal immune infiltrates by immunohistochemistry demonstrated the presence of both T and B cells (Fig 5D). Interestingly, the overall steady-state levels of OXPHOS proteins and OXPHOS complexes did not change in kidneys of old (55 weeks) *Mgme1*<sup>-/-</sup> mice (S4A–S4C Fig), but COX deficient cells were demonstrated by

Nitrotetrazolium blue exclusion assay (NBTx) enzyme histochemistry (S4D Fig) in kidney sections from four of five analysed mutant animals. These results show that loss of MGME1 causes glomerular changes and kidney inflammation ultimately leading to severe chronic progressive nephropathy and possible death due to renal failure.

## Discussion

In this study we report a detailed phenotypic characterization of mice lacking the mitochondrial exonuclease MGME1. Despite having clear aberrations in mtDNA, young *Mgme1* knockout animals are healthy and fertile without apparent morphological alterations [13]. However, weight loss and functional changes of multiple organs become apparent as the animals age. Lack of the MGME1 protein alters body composition and causes a reduction in fat mass. There are also pathological changes of the lens and the retina. Most importantly, mutant animals develop a severe kidney pathology with glomerular changes, tubulointerstitial fibrosis and inflammation ultimately leading to a nephrotic syndrome coupled with reduced life span.

The mtDNA maintenance diseases are genetically and clinically heterogenous with severities ranging from infantile fatal forms to adult onset of mild disease. Whereas organs with high energy demands, such as heart, skeletal muscle and the central nervous system, often are affected, manifestations can occur in many different organs [19]. Kidneys contain a high density of mitochondria, particularly in the cortical tubules, but all sections of the nephron may be affected by mitochondrial dysfunction. Proximal tubulopathy (Fanconi syndrome) is a common renal phenotype frequently seen in early-onset mitochondrial disease patients, whereas glomerular disease is more frequent in adults [20]. Human patients with MGME1 deficiency were reported to have an adult onset, multisystem mitochondrial disorder including progressive external ophthalmoplegia (PEO), muscle wasting, muscle weakness, exercise intolerance, cerebellar atrophy, cerebellar ataxia and cardiomyopathy [9]. With regard to kidney-related phenotypes, those patients were reported to develop nephrolithiasis [9]. Also, mitochondrial dysfunction has been linked to chronic kidney disease in humans [21]. Interestingly, some end-stage alterations in human chronic kidney disease are similar to those in rodents, e.g. proteinuria, tubular atrophy, scarring, as well as increases of blood urea nitrogen (BUN) and serum creatinine levels [22].

MGME1 deficiency in patients and mice leads to high levels of a linear deleted mtDNA fragment in many different tissues, but the involved mechanisms are much debated. To address this question, we used *Mgme1*[-/-] knockout mice [13] to study linear deletion formation. Our data argue against a proposed model where MGME1 functions as a key enzyme that degrades linear mtDNA fragments [14]. Firstly, our data show that deleted linear mtDNA fragments do not accumulate with age in MGME1 knockout mice. Secondly, *in organello* mtDNA replication assays show that there is a substantial *de novo* formation of linear deleted mtDNA in the absence of MGME1, whereas the degradation of this linear fragment is not affected. Two additional replication factors, POLγA and the replicative mtDNA helicase TWINKLE have been suggested to act together with MGME1 to degrade linear mtDNA fragments [14,16,17]. However, as POLγA possesses a 3′-5′exonuclease activity and MGME1 preferentially degrades single-stranded DNA in the 5′-3′ direction [9], it seems unlikely that any POLγA can compensate for loss of MGME1. Instead, the results in our study support a previously proposed model where MGME1 removes flaps generated when mtDNA replication is reaching completion [23]. In the absence of MGME1, these flaps will persist and prevent the ligation step necessary to finalize mtDNA replication, which will result in the formation of linear deleted mtDNA [23].

Unexpectedly, the *Mgme1*[-/-] mouse model also links defective mtDNA replication to inflammatory disease manifestations. Ageing *Mgme1*[-/-] mice develop kidney inflammation, tubulointerstitial fibrosis and glomerular changes leading to nephrotic syndrome. It is unclear at this point how defective mtDNA replication triggers age-associated inflammation. An increasing literature indicates that mitochondria are key participants in innate immune pathways, representing signaling platforms and participating in effector responses [24]. Under certain pathological conditions various mitochondrial ligands or damage-associated molecular patterns (DAMPs), including mtDNA, can be released from mitochondria and recognized by different pattern recognition receptors (PRRs). It is now appreciated that mtDNA can stimulate different PRRs, including cytosolic cGAS, endosomal localised TLR9 and inflamasomes to activate various pro-inflammatory signaling pathways [25–27]. Future studies should aim to investigate if the inflammation associated with MGME1 defficiency is triggered by mtDNA release.

## Materials and methods

### Ethics statement

This study was performed in strict accordance with the recommendations and guidelinesof the Federation of European Laboratory Animal Science Associations (FELASA). The protocol was approved by the "Landesamt für Natur, Umwelt und Verbraucherschutz Nordrhein- Westfalen" (reference numbers 81.02.04.2020.A082, 84–02.04.2015.A103 and 84–02.50.15.004) and by the Stockholm ethical committee (Stockholms djurförsöksetiska nämnd) under the ethical permit 1206–2019. The mice were maintained according to the GMC housing conditions (www.mouseclinic.de) in strict accordance with directive 2010/63/EU, the local government and German laws. The GMC holds a general license to run phenotype assessments in mice and all tests are approved by the responsible authority of the district government of Upper Bavaria.

### Animals and housing

This study was performed in strict accordance with the recommendations and guidelinesof the Federation of European Laboratory Animal Science Associations (FELASA). The protocol was approved by the "Landesamt für Natur, Umwelt und Verbraucherschutz Nordrhein- Westfalen" (reference numbers 81.02.04.2020.A082, 84–02.04.2015.A103 and 84–02.50.15.004). The *Mgme1* knockout and wild-type mice on a C57BL/6N background were housed in standard individually ventilated cages (45 x 29 x 12 cm) under a 12h light/ dark schedule in controlled environmental conditions of 22 ± 2°C and 50 + 10% relative humidity and fed a normal chow diet and water *ad libitum*. Generation of the *Mgme1* knockout mice was described before [13]. Characterization on 14 female *Mgme1*[-/-] and 15 *Mgme1*[+/+] littermate controls and 15 male *Mgme1*[-/-] and 15 *Mgme1*[+/+] littermate controls was performed at the German Mouse Clinic. Body weight measurements started at the age of 8 weeks, the phenotyping examination of 3 female *Mgme1*[-/-] and 4 females controls, 5 male *Mgme1*[-/-] and 6 male controls was performed at 68 weeks of age, comprising tests examining Neurology, Dysmorphology, Metabolism, Cardiology, Clinical Chemistry and Pathology. The mice were maintained according to the GMC housing conditions (www.mouseclinic.de) in strict accordance with directive 2010/63/EU, the local government and German laws. The GMC holds a general license to run phenotype assessments in mice and all tests are approved by the responsible authority of the district government of Upper Bavaria. The phenotypic tests were performed as outlined in the standard operating procedures (SOP) linked to the EMPReSS website http://empress.har.mrc.ac.uk.

Experimental groups were assigned according to the genotype of the animals. The selection of the mice for testing was balanced, control and mutants were measured alternately. Most of the tests were not conducted in blinded conditions because the results were recorded directly

by the machines and, therefore, not influenceable by the examiner. The experiment was conducted in blinded conditions whenever there could have been an influence from the investigator. All the procedures are described in SOPs. Metadata for each data point was recorded throughout the measurements.

## Body composition assessment (GMC)

Body composition was analysed with time domain-nuclear magnetic resonance (Bruker Minispec LF 50) in live mice without the administration of anesthesia at 68 weeks of age of 4 wild-type and 3 mutant females and 6 wild-type and 5 mutant males. Body weight measured at the same time of the analysis was used to determine the body fat and lean percent.

## Pathological analyses and immunohistochemistry (GMC)

Microscopy and histopathological analyses using hematoxylin and eosin (H&E) staining on formalin-fixed paraffin-embedded sections (3 μm) were performed as described in www.mouseclinic.de/screens/pathology. A Leica Bond III (Leica Biosystems) automatic stainer was used for immunohistochemistry. Heat-induced antigen retrieval was performed with citrate buffer (pH 6) for 30 minutes (AR9961; Bond Epitope Retrieval Solution; Leica Biosystems). Antibodies against CD3 (Clone SP7; ZYT-RBG024; Zytomed systems) and CD45R/B220 (Clone RA3-6B2; 550286; BD Pharmingen) were employed and the staining was detected with DAB chromogen. PAS (Periodic acid-Schiff) staining was performed to study glomerular changes in the kidneys (in particular mesangial expansion). Tubulointerstitial fibrosis was assessed with Sirius Red staining using standard protocols. The slides were scanned using a Hamamatsu NanoZoomer 2.0HT digital scanner and analyzed by two independent pathologists using NDP.view2 software (Hamamatsu Photonics).

Eye histology: At sacrifice the eyes were enucleated and after 24 hour Davidson fixation were embedded in Technovit 8100 (Heraeus Kulzer, Wehrheim, Germany) and kept for polymerization for 6–10 hours at 4˚C. Samples, cut in 2 μm sagitally through the middle of the eye ball, were stained with basic fuchsin and methylene blue. Slides were scanned (NanoZoomer 2.0HT Digital slide scanner, Hamamatsu, Japan) and taken images were processed with an image-processing program (Adobe Inc., 2019. Adobe Illustrator).

## Blood collection

Blood samples were collected under isoflurane anaesthesia by retrobulbar puncture as a final blood withdrawal without prior fasting of the animals from 5 homozygous mutant and 6 wild-type males as well as 3 homozygous and 4 wild-type females. An aliquot of 50μl whole blood was collected in EDTA-coated end-to-end capillaries and diluted 1:5 with buffer provided from Sysmex (Cell-Pack buffer) for subsequent analysis of basic haematological parameters. Blood samples for clinical chemistry analyses were collected in Li-heparin-coated tubes and stored at room temperature until centrifugation (4500xg, 10 min) and separation of plasma aliquots for further analyses. Plasma samples were frozen at -80˚C until analysis within one week after collection.

## Clinical chemistry and cytokines

For the clinical chemistry analyses plasma samples were thawed at room temperature, diluted 1:2 with deionised water, mixed thoroughly, and centrifuged again (5000xg, 10 min) to remove clots from the sample. Measurement of circulating biochemical parameters was performed using a clinical chemistry analyser (Beckman Coulter AU 480 autoanalyzer, Krefeld,

Germany). A broad set of parameters was measured using the respective kits provided by Beckman Coulter, in order to determine various enzyme activities as well as plasma concentrations of specific substrates and electrolytes in *ad libitum* fed mice [28]. Multiplex measurement of the proinflammatory cytokines IL-6, TNF-α, KC/GRO, IL-2, IL-10, IFN-γ, IL-5, IL-4 and IL-1b was performed in plasma of a separate cohort of mice at x weeks of age (wt n = 11 ; Mgme$^{-/-}$ n = 20).

### Nitrotetrazolium blue exclusion (NBTx) staining assay

Tissue Preparation—kidneys were quickly frozen in 2-methylbutane, in a glass beaker cooled by immersion in liquid nitrogen. Frozen tissues were stored at -80˚C until ready to use. Thin sections of 10 μm were cut with a cryostat at -20˚C (OFT 5000, Bright Instruments, Luton, UK) and mounted on Superfrost Plus microscope slides (Menzel, Thermo Scientific, Waltham, MA, USA) and air-dried for 5 to 10 min. Slides were kept at -80˚C for maximum a few months to avoid the loss of enzyme activity.

Staining Protocol—slides were taken out of the -80˚C freezer and thawed briefly at room temperature on a slide holder without lid. Sections (3 sections per slide) were then covered with 1ml PBS for 10 min in an incubator set at 21˚C (1 ml per slide). PBS was discarded and replaced with 1 ml NBTx solution. Sections of kidneys were left 30 min at 21˚C in the incubator and washed briefly in purified water followed by dehydration in ethanol (2 min in 50%, 75%, 96%, 100% followed by an extra 5 min in 100% ethanol). Finally, slides were immersed for 5 min in two changes of xylene before mounted on coverslips with Cytoseal (Thermo Scientific, Darmstadt, Germany).

### Isolation of mitochondria from mouse tissues

Mitochondria were isolated from mouse tissues using differential centrifugation as previously described [29]. Briefly, freshly obtained tissues were cut, washed with ice cold PBS and homogenized in mitochondrial isolation buffer containing (320 mM sucrose, 10 mM Tris/HCl pH 7.4, and 1 mM EDTA) supplemented with 1× Complete protease inhibitor cocktail (Roche) by using a Teflon pestle (Schuett Biotec). After 10 min centrifugation at 1000×g using swing-out rotor at 4˚C the supernatants were subsequently spun at 10000×$g$ for 10 min at 4˚C to isolate the mitochondria.

### mtDNA extraction and Southern blot analysis

Total DNA or mtDNA was isolated from pulverized tissue or purified mitochondria respectively, using Gentra Puregene Tissue Kit (Qiagen) according to kit instructions. DNA quantification was performed with the Qubit 1.0 fluorometer (Thermofisher). 300–1000 μg of DNA were digested with *SacI* restriction nuclease and DNA fragments were separated by agarose gel electrophoresis, transferred to nitrocellulose membranes (Hybond-N+ membranes, GE Healthcare) and hybridized with αP$^{32}$-dCTP-labeled probes. For 7S DNA, Southern blot samples were heated for 3 min at 93˚C prior to loading. Southern blot signals were quantified using MultiGauge or ImageJ softwares.

### In organello replication

1 mg of freshly isolated heart mitochondria were resuspended in 0.5 ml of incubation buffer (25 mM sucrose, 75 mM sorbitol, 100 mM KCl, 10 mM K$_2$HPO$_4$, 0.05 mM EDTA, 5 mM MgCl$_2$, 1 mM ADP, 10 mM glutamate, 2.5 mM malate, 10 mM Tris–HCl, pH 7.4, 1 mg/ml fatty acid-free bovine serum albumin, 50 μM each of dTTP, dCTP and dGTP and 20 μCi

α-$^{32}$P-dATP (3000 Ci/mmol). Incubation was carried out at 37˚C for 2h on a rotating wheel. For the chase reisolated mitochondria were incubated in 0.5 ml of incubation buffer supplemented with all four non-radiolabeled dNTPs (50 μM) for indicated time. After incubation, mitochondria were pelleted at 9000 rpm for 4 min and washed twice with washing buffer (10% glycerol, 10 mM Tris–HCl, pH 6.8, 0.15 mM MgCl$_2$). In the following step DNA isolation and Southern blot analysis were performed as described above.

### Western blot analysis and BN-PAGE

20 μg of isolated mitochondria were resuspended in 4X Lämmli-Buffer (4% SDS, 20% Glycerol, 120mM Tris, 0,02% Bromophenol Blue), proteins were separated on 4–12% NuPAGE gels (Invitrogen) and transferred on Hybond-P membrane (GE Helthcare). MitoProfile total OXPHOS antibody cocktail (MitoSciences) antibody was used for the western blotting. Western blot signals were quantified using the ImageJ processing program.

BN-PAGE and subsequent in gel activity were performed as previously described [30]. For BN-PAGE, 75 μg of isolated kidney mitochondria were lysed in 50 μl solubilization buffer (20 mM Tris pH 7.4; 0.1 mM EDTA; 50 mM NaCl; 10% [v/v] glycerol) containing 1% (w/v) digitonin (Calbiochem) and mixed with loading dye (5% [w/v] Coomassie Brilliant Blue G-250, 150 mM Bis-Tris, and 500 mM ε-amino-n-caproic acid [pH 7.0]). BN-PAGE samples were resolved on self-made 3%– 13% gels. Protein complexes were visualized using *in gel* activity staining for complexes I, II and IV. For CI *in gel* activity the BN-PAGE gel was incubated in 2 mM Tris/HCl pH 7.4, 0.1 mg/ml NADH (Roche) and 2.5 mg ml$^{-1}$ iodonitrozolium (Sigma) for about 10 minutes. *In gel* CIV activity was determined by incubating the BN-PAGE gels in 10 ml of 0.05 mM phosphate buffer pH 7.4, 25 mg 3.3′-diamidobenzidine tetrahydrochloride (DAB), 50 mg Cyt *c*, 3.75 g Sucrose and 1 mg Catalase for approximately 1h. For the CII assay, the buffer contained 200 μl of sodium succinate (1 M), 8 μl of phenazine methosulfate (250 mM dissolved in DMSO), and 25 mg of NTB in 10 ml of 5 mM Tris/HCl, pH 7.4. Incubation of 10–30 min was required. All in gel staining reactions were carried out at room temperature and stopped using solution containing 50% methanol, 10% acetic acid for 30 min.

### Statistics

Tests for genotype effects of the phenotyping data were made by Wilcoxon rank sum test for parametric data, if not indicated otherwise. A P < 0.05 has been used as a level of significance; a correction for multiple testing has not been performed. Figures were prepared using GraphPad Prism version 7.00 for Windows (GraphPad Software, La Jolla, California, USA).

### Supporting information

**S1 Fig. Gross phenotypes.** (A) Survival percent by sex. The attrition rate in females was significant with a p = 0,0017 Log-rank (Mantel-Cox) test. 11 from 14 female mutants and 2 from 15 control, 8 from 13 males mutants and 4 from 15 control animals did not reach the end of phenotyping. (B) Lean mass indicated as percent of body weight and adiposity index intended as ratio between fat and lean mass of 68 week-old n = 4 wild-type and 3 *Mgme1*$^{-/-}$ females and 6 wild-type and 5 *Mgme1*$^{-/-}$ males. Fat mass is indicated as percent of bodyweight. Values are given as mean ± SD, $^{**}$ P ≤ 0.01, Unpaired Mann Whitney test.
(TIF)

**S2 Fig. Southern blots of mtDNA isolated from various tissues of young and old control and *Mgme1*$^{-/-}$ mice that were used for the quantification presented in Fig 2A.** (A) liver, (B)

kidney, (C) heart, (D) skeletal muscle and (E) brain.
(TIF)

**S3 Fig. Southern blots of young and old control and *Mgme1*<sup>-/-</sup> mice that were used for the quantification presented in Fig 2C and 2D.** (A) heart (B) kidney. Panels on the left were used for quantification of total and full length mtDNA levels and the panels on the right for quantification of 7S DNA. The 18S rDNA was used as a loading control.
(TIF)

**S4 Fig. Cell specific OXPHOS dysfunction in aged kidney of *Mgme1*<sup>-/-</sup> mice.** (A) Steady-state levels of OXPHOS subunits in heart and kidney of control (+/+) and Mgme1 knockout (-/-) mice. (B) Quantification of steady state levels of OXPHOS subunits from kidney tissue. (C) BN-PAGE analysis followed by in-gel enzyme activities of complexes I, IV and II (loading control) in Mgme1 knockout (-/-) and wild-type (+/+) kidney mitochondria at 55 weeks of age. (D) NBTx staining of kidney sections from Mgme1 knockout (-/-) and wild-type (+/+) mice. Representative pictures from multiple analysed sections from 5 wild-type (+/+) and 5 knockout (-/-) animals are shown.
(TIF)

**S1 Table. Late adult screening pipeline overview.** In orange: Modified protocol for the Mgme1 knockout mice and controls.
(PDF)

**S2 Table. Sumarized results of the clinical chemistry analysis.**
(PDF)

**S1 Data. The numerical data underlying the graphs or summary statistics in this study.**
(XLSX)

## Acknowledgments

We are grateful to Petra Kirschner for expert technical assistance.We thank the FACS & Imaging core facility of Max Planck Institute for biology of Ageing in Cologne.

## Author Contributions

**Conceptualization:** Dusanka Milenkovic, Eckhard Wolf, Helmut Fuchs, Valerie Gailus-Durner, Martin Hrabě de Angelis, Nils-Göran Larsson.

**Data curation:** Dusanka Milenkovic, Adrián Sanz-Moreno, Julia Calzada-Wack, Birgit Rathkolb, Oana Veronica Amarie, Raffaele Gerlini, Antonio Aguilar-Pimentel, Jelena Misic, Marie-Lune Simard.

**Formal analysis:** Dusanka Milenkovic, Adrián Sanz-Moreno, Julia Calzada-Wack, Birgit Rathkolb, Oana Veronica Amarie, Raffaele Gerlini, Antonio Aguilar-Pimentel, Jelena Misic.

**Funding acquisition:** Martin Hrabě de Angelis, Nils-Göran Larsson.

**Investigation:** Dusanka Milenkovic.

**Methodology:** Dusanka Milenkovic, Adrián Sanz-Moreno, Julia Calzada-Wack, Birgit Rathkolb, Oana Veronica Amarie, Raffaele Gerlini, Antonio Aguilar-Pimentel, Jelena Misic, Marie-Lune Simard.

**Resources:** Nils-Göran Larsson.

**Supervision:** Eckhard Wolf, Helmut Fuchs, Valerie Gailus-Durner, Nils-Göran Larsson.

**Validation:** Jelena Misic.

**Visualization:** Adrián Sanz-Moreno, Julia Calzada-Wack, Birgit Rathkolb, Oana Veronica Amarie, Raffaele Gerlini, Antonio Aguilar-Pimentel, Jelena Misic.

**Writing – original draft:** Dusanka Milenkovic, Nils-Göran Larsson.

**Writing – review & editing:** Dusanka Milenkovic, Nils-Göran Larsson.

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
