## [Decision Letter · Decision Letter 0]

13 Jan 2022

Dear Dr Larsson,

Thank you very much for submitting your Research Article entitled 'Mice lacking the mitochondrial exonuclease MGME1 develop inflammatory kidney disease with glomerular dysfunction' to PLOS Genetics.

The manuscript was fully evaluated at the editorial level and by independent peer reviewers. The reviewers appreciated the attention to an important problem, but raised some concerns about the current manuscript. Based on the reviews, we will not be able to accept this version of the manuscript, but we would be willing to review a revised version. We cannot, of course, promise publication at that time.

If you decide to revise the manuscript for further consideration at PLOS Genetics, please aim to resubmit within the next 60 days, unless it will take extra time to address the concerns of the reviewers, in which case we would appreciate an expected resubmission date by email to plosgenetics@plos.org.

[LINK]

Please do not hesitate to contact us if you have any concerns or questions.

Yours sincerely,

Carlos T. Moraes

Guest Editor

PLOS Genetics

Gregory Barsh

Editor-in-Chief

PLOS Genetics

Dear Dr. Larsson,

Please find attached the Reviewers' comments. They found the study to be of high quality and interesting but converged in asking more information about the levels of mtDNA (linear and total) at different ages. Southern blots may be required to address this issue. They also raised a few more issues that need to be addressed.

Reviewer's Responses to Questions

**Comments to the Authors:**

Reviewer #1: The manuscript by Milenkovic et al. characterizes the aging phenotype of mice lacking MGME1 and shows that these mice present with body weight loss, retinopathy, a kidney inflammation consistent with nephrotic syndrome, leading to premature death. The same group has previously shown that young mice lacking MGME1 show abnormal mtDNA replication, that leads to accumulation of linear mtDNA fragments (in manuscript citation [13]). In this manuscript, comparison of the relative content of linear mtDNA fragments between young (10 weeks) and old (50 weeks) animals showed that MGME1 is not necessary for the degradation of the linear mtDNA fragments, as no accumulation during aging was detected. To prove, that linear mtDNA fragments are indeed degraded in MGME1-lacking mice, in organello pulse-chase experiment in isolated heart mitochondria was performed. The manuscript is well written and data are clearly presented.

Questions:

Was there a decrease in the mtDNA levels in aged animals compared to the younger ones?

Was there an accumulation of the 7S DNA in the aged mice compared to younger mice? Determination of 7S DNA levels in aged vs. young mice would by the same logic as for the linear mtDNA fragments show if MGME1 is responsible for the degradation of this DNA, as it was suggested previously?

There is an increase in the total amount of newly synthetized mtDNA in MGME1-/- mice in the in organello pulse/chase experiment, which seems to be rapidly degraded during the chase period. Is there a faster degradation of the newly synthetized mtDNA in MGME1-/- mice compared to controls and does this lead to the mtDNA depletion, rather than a problem with replication?

Were any inflammatory factors (increase in IL6/TNF/cytokines) detected in the aged mice, as Table S2 indicates that Immunology and Allergy module was performed?

Minor comments:

- Please use decimal point in P-values in the paragraph “MGME1 deficiency causes kidney inflammation and nephropathy”

- Abstract: “formation of a linear deleted mtDNA fragments” – delete “a”

Reviewer #2: This manuscript by Milenkovic et al, expands upon the understanding of the role of the mitochondrial genome maintenance exonuclease (MGME1) in vivo. In a previous study the authors examined the role of MGME1 at early ages, but this study utilized aged knockout mice to analyze the function of MGME1 and the associated disease pathophysiology, as MGME1 mutations generally manifest as disease later in life in patients.

While aging the MGME1 knockouts it was found that these animals died prematurely, so doing the standard late-adult phenotyping (at 71 weeks) was not possible. Instead, the extensive phenotyping analyses were performed at 50 weeks. This modified “late-adult” phenotyping protocol also included neurological and morphological examinations, cardiovascular monitoring, and body composition measurements to give a complete and extensive assessment of the model. From these studies the authors concluded that striking phenotypes of the aged MGME1 knockout mice included reduced weight gain during aging and later a marked weight loss. Additionally, the MGME1 knockout mice developed cataracts due to abnormal fiber rearrangement and infiltrating epithelial cells in the lens. Retinopathy as also identified, where the overall thickness of the retina and retinal layers was reduced even though all the layers of the retina were present. Renal failure was observed through elevated plasma urea and creatinine levels. The nephrotic syndrome associated with renal failure was further confirmed through histological assessments of the kidney glomeruli. Furthermore, there was detection of infiltrating T and B cells in the kidneys.

This conclusions from this manuscript are novel and compelling for the mitochondrial biology field and the overall understanding of how defective mtDNA replication can lead to manifestations of inflammatory disease. The authors show evidence that contradicts the previous proposal where MGME1 plays a role in the degradation of linear mtDNA, and instead assert that MGME1 plays a crucial role in the completion of mtDNA replication. From the phenotyping experiments the authors assert that the loss of MGME1 leads to a dramatic phenotype that includes progressive weight loss, retinopathy, cataracts, and nephrotic disease. However, there are some lingering questions that arise, particularly relating to how the MGME1 knockout mouse compares to other models of mitochondrial DNA instability or defective replication that could be discussed in the text.

1) One of the major conclusions the authors draw from their studies is that MGME1 does not play a role in degrading linear mtDNA fragments, as proposed by Peeva et al (2018). However, that paper and the papers by Medeiros et al (2018) and Nissanka et al (2018), also show that TWNK and POLG can also play a role in degrading linear mtDNA fragments. Can the authors comments on why they think these other proteins do not compensate for the loss of MGME1 to degrade linear mtDNA?

2) In Figure 2 the authors show that the levels of the linear deletion are not significantly altered between young and aged MGME1 knockout animals in different tissues. Are the levels of total mitochondrial DNA altered between young and aged MGME1 knockout animals?

3) With the conclusion that the impaired mitochondrial DNA replication is directly responsible for the kidney inflammation, could to authors comment on how their studies with the MGME1 knockout mouse compare to other mouse models of impaired mitochondrial DNA replication?

4) One of the hallmarks of the MGME1 knockout mouse model is the formation of the linear mtDNA deletion, which is also seen in the POLG mutator mouse. Comparatively however, the POLG mutator mouse does not have the same effect on mtDNA replication as the MGME1 knockout. Can the authors discuss on if they think that these linear deletions cause the systemic inflammation that causes the progressive kidney disease only in the MGME1 knockouts compared to the POLG mutator?

Reviewer #3: Mice Lacking the Mitochondrial Exonuclease MGME1 Develop Inflammatory Kidney Disease with Glomerular Dysfunction

Summary: The current publication reports on the age-associated pathology in Mgme1 KO mice and aims to expand on the role of MGME1.

The most impressive part of this manuscript is the amount of detail given to the mouse studies. It is evident that the parameters to arrive at these observational findings were meticulously planned, and there are vast amounts of data that were collected from these animals. The authors focused their attention on some of the striking pathophysiological features of these mice. Namely: survival, weight loss, body mass composition, blood chemistry and gross histology. In addition, they tackle the ongoing controversy in the field regarding the role of MGME1 in degrading linear deleted mtDNA.

Regarding the observational aspect of this publication the following points may be worth noting:

• In figure 3, where the author reports the gross eye histological images, a close-up section of the WT lens isn’t provided to have a full comparison to a KO counterpart. At the same time, the author signals that there is an invagination visible in the KO that to the untrained eye looks like tissue processing artifact. This same artifact can be seen in pane A just right of center towards the top of the image. Is this also a tissue invagination? Are there more in the KO? If perhaps it is an artifact perhaps highlighting may not serve to highlight the differences between WT and KO. The differences in the lens epithelium are evident enough. And if the differences lie in quantities, then this should be stated.

• In figure 1 and supplemental figure 1, there is a decrease in fat mass (%BW) and conversely there is an increase in lean mass (%BW). This is interesting considering MGME1 human pathology where muscle wasting has been reported. Although logically, both muscles and fat must be wasting in these animals, perhaps a mention or presentation of a graphic showing the decrease in lean muscle mass across time can further provide support for how this model is like human disease, as is the presented, it shows an increase in lean muscle mass (as a percentage of total body weight).

• In supplemental figure 2, the author reports that there is no apparent COX deficiency or OXPHOS deficiency in Mgme1 KO animals, however only the blot and BN-PAGE gel is shown. Consider quantifying these data and drawing statistics to show this very same conclusion as it is a subsection in the manuscript and not a point in one of the sections. Additionally, consider mentioning how the NBTx staining of the kidney sections showing COX deficient cells fits into the greater picture of kidney physiology. Perhaps, determining the identity of these cells (if possible) may provide a clearer picture of how kidney disease comes about and an explanation to what appears to be incongruous findings.

When it comes to the experimental aspect of the manuscript, the author claims that the findings in this manuscript contradict a previous publication that states that MGME1 degrades long linear mtDNA molecules. In the current manuscript, Mgme1 KO animals are associated with de novo formation of linear fragments that are constantly made and degraded. This is only partially substantiated by the experimental data presented in figure 2. Not only does the pulse chase experiment have no signal for the WT samples, but there is no control for what typical linear fragment degradation (in WT cells) would look like in this experiment. Although WT cells have no linear deleted mtDNA, the author states that the degradation of linear fragments in these Mgme1 KO mice is not affected by the absence of MGME1. What if the absence only dampens degradation without full ablation? Without a comparative linear deleted fragment degradation example in WT mice this conclusion cannot be drawn. What can be gathered is that the linear deleted fragment is degraded, and that perhaps there is greater mtDNA turnover in the Mgme1 KO cells which may lead to production of this fragment. If possible, providing more experimental data about mtDNA turnover in this set of mice 1) may demonstrate the differences in mtDNA quantity between the two groups and 2) may elucidate whether the production of these linear mtDNA fragments is directly tied to mtDNA replication. An interesting point seeing as MGME1 is involved in post replication processing of mtDNA.

Overall, this is a strong observational manuscript with meticulous attention placed on data collection and analysis. The conclusions drawn surrounding the role of MGME1 and linear deleted mtDNA lack additional experimental evidence. However, the conclusions drawn from the experiments presented can be adjusted if further experimental evidence cannot be provided to move forward with the current claim.

**Have all data underlying the figures and results presented in the manuscript been provided?**

Reviewer #1: **No: **There are no examples of the Southern blots used for the quantification of the linear mtDNA fragments in the Supporting information.

Reviewer #2: Yes

Reviewer #3: Yes

PLOS authors have the option to publish the peer review history of their article (what does this mean?). If published, this will include your full peer review and any attached files.

Reviewer #1: No

Reviewer #2: No

Reviewer #3: No

---

## [Decision Letter · Decision Letter 1]

5 Apr 2022

Dear Dr Larsson,

We are pleased to inform you that your manuscript entitled "Mice lacking the mitochondrial exonuclease MGME1 develop inflammatory kidney disease with glomerular dysfunction" has been editorially accepted for publication in PLOS Genetics. Congratulations!

Yours sincerely,

Carlos T. Moraes

Guest Editor

PLOS Genetics

Gregory Barsh

Editor-in-Chief

PLOS Genetics

Comments from the reviewers (if applicable):

The authors have addressed all the concerns. The Reviewers had only minor corrections.

Reviewer's Responses to Questions

**Comments to the Authors:**

Reviewer #1: The authors have adequately answered all my questions.

Minor comments:

Fig. 2A – please specify which bar corresponds to young and which to old animals in the figure/figure legend.

Reviewer #2: The authors have responded to all my comments, and I have no further scientific concerns, though have noted a few additional grammatical errors that should be addressed.

Minor grammatical comments:

1) In the author summary “firbrosis” should be “fibrosis”

2) In the second to last sentence of the MGME 1 deficiency results in body weight decline section of the RESULTS, the figure reference should be (Fig 2C and D, S3A and B Fig) there is an unneeded “_” between “S3A and B”

3) In the first sentence of the MGME 1 deficiency causes inflammation and nephropathy section of the RESULTS, the cytokines should be TNF-alpha and INF-gamma (switch to symbols instead of words, rather than the letters "a" and "g")

**Have all data underlying the figures and results presented in the manuscript been provided?**

Reviewer #1: Yes

Reviewer #2: Yes

PLOS authors have the option to publish the peer review history of their article (what does this mean?). If published, this will include your full peer review and any attached files.

Reviewer #1: No

Reviewer #2: No

**Data Deposition**

http://datadryad.org/submit?journalID=pgenetics&manu=PGENETICS-D-21-01644R1

**Press Queries**

---

## [Editor Report · Acceptance letter]

5 May 2022

PGENETICS-D-21-01644R1 

Mice lacking the mitochondrial exonuclease MGME1 develop inflammatory kidney disease with glomerular dysfunction 

Dear Dr Larsson, 

We are pleased to inform you that your manuscript entitled "Mice lacking the mitochondrial exonuclease MGME1 develop inflammatory kidney disease with glomerular dysfunction" has been formally accepted for publication in PLOS Genetics! Your manuscript is now with our production department and you will be notified of the publication date in due course.

With kind regards,

Agnes Pap

PLOS Genetics

On behalf of:
